# Preparation and Characterization of Thermoresponsive Poly(*N*-Isopropylacrylamide) for Cell Culture Applications

**DOI:** 10.3390/polym12020389

**Published:** 2020-02-09

**Authors:** Lei Yang, Xiaoguang Fan, Jing Zhang, Jia Ju

**Affiliations:** 1College of Chemistry, Chemical Engineering and Environmental Engineering, Liaoning Shihua University, Fushun 113001, China; 66zj@163.com (J.Z.); jujia@lnpu.edu.cn (J.J.); 2College of Engineering, Shenyang Agricultural University, Shenyang 110866, China

**Keywords:** poly(*N*-isopropylacrylamide), thermosensitive polymer, thermoresponsive culture platform, cell culture, polymer characterization

## Abstract

Poly(*N*-isopropylacrylamide) (PNIPAAm) is a typical thermoresponsive polymer used widely and studied deeply in smart materials, which is attractive and valuable owing to its reversible and remote “on–off” behavior adjusted by temperature variation. PNIPAAm usually exhibits opposite solubility or wettability across lower critical solution temperature (LCST), and it is readily functionalized making it available in extensive applications. Cell culture is one of the most prospective and representative applications. Active attachment and spontaneous detachment of targeted cells are easily tunable by surface wettability changes and volume phase transitions of PNIPAAm modified substrates with respect to ambient temperature. The thermoresponsive culture platforms and matching thermal-liftoff method can effectively substitute for the traditional cell harvesting ways like enzymatic hydrolysis and mechanical scraping, and will improve the stable and high quality of recovered cells. Therefore, the establishment and detection on PNIPAAm based culture systems are of particular importance. This review covers the important developments and recommendations for future work of the preparation and characterization of temperature-responsive substrates based on PNIPAAm and analogues for cell culture applications.

## 1. Introduction

Poly(*N*-isopropylacrylamide) (PNIPAAm) is a generally known thermoresponsive polymer that can be synthesized from monomers through free radical polymerization, and it is easily functionalized making it available in wide fields. PNIPAAm is very remarkable since it shows invertible changes in hydrated and dehydrated forms during temperature variation, and displays lower critical solution temperature (LCST) at around 31 °C in aqueous solutions, which is approaching the physiological threshold [1,2,3]. The thermoresponsive behavior of PNIPAAm benefits from the contribution of its special chemical structure, as displayed schematically in Figure 1. The molecular chains of PNIPAAm are composed of hydrophilic amide linkages and hydrophobic isopropyl groups. When the environmental temperature is below LCST, the hydrogen bonds between water molecules and amide linkages urges molecular chains to form stable hydration structure surrounding isopropyl groups. When the ambient temperature goes up, the hydration structure breaks down and the nonhydrated hydrogen bonding between amide groups and the hydrophobic interactions derived from isopropyl groups play predominant roles, which force the water molecules initially bound to PNIPAAm to spread outwards.

PNIPAAm has been used for smart materials, which mainly utilizes surface property changes and volume phase transitions with response to ambient temperature. The versatility of PNIPAAm has led to finding uses in cell culture [4], tissue engineering [5], enzymatic immobilization [6], drug delivery [7], wound dressing [8], biosensors [9], etc. One of the most important applications of PNIPAAm is cell culture. Efficient recovery of adherent cells from culture surfaces becomes essential to conduct cell manipulation in vitro. There is scientific evidence to prove that the common treatments like enzymatic hydrolysis and mechanical scraping are harmful to membrane proteins and further cell functions [10,11]. To provide therapeutic cells with high quality for biomedical technology, it is generally admissible to recover cells from culture substrates by means of non-enzymatic and non-mechanical methods. The development of temperature-responsive substrates is expectedly helpful to replace the traditional harvesting methods. The precise and mild control over cell attachment and detachment is achieved by use of surface wettability changes and/or bulk structure variations of PNIPAAm mediated culture substrates with respect to temperature shift. When cooled to below LCST, the adherent cells spontaneously separate from thermoresponsive surfaces along with integral membrane proteins and most intercellular linkage proteins, which is beneficial in protecting the ultrastructure and functions of targeted cells [12,13]. This review focuses on the preparation and characterization of PNIPAAm modified substrates for cell culture applications. It firstly and systematically describes many useful fabrication processes for temperature-responsive culture substrates, and points out their respective advantage, disadvantage and adaptive conditions. Secondly, it gives principal statement of characterization techniques aimed at the above-mentioned thermoresponsive substrates. Finally, it is concluded with recommendations concerning future work that is needed to address the preparation and characterization of PNIPAAm based substrates for cell culture applications.

## 2. Preparation of PNIPAAm Medicated Substrates for Cell Culture

For cell culture applications, the thermoresponsive products involved in PNIPAAm and its derivatives are generally classified into planar films and spatial supports. There are currently many ways to prepare them.

### 2.1. Planar Films

The studies on the majority of innovative materials used for cell culture mainly start from two-dimensional (2D) systems, which are considered as fundamental research or preliminary exploration. There are basically three different strategies for preparation of planar films, grafting, coating or both together, as illustrated in Figure 2.

#### 2.1.1. Covalent Grafting Films

For insoluble grafting films, it is imperative to immobilize PNIPAAm and derivatives to the designated substrates (such as glass, silicon, polystyrene, polypropylene, etc.). The procedures usually require sophisticated and expensive processing means [14,15,16], such as electron beam (EB) irradiation, plasma treatment, ultraviolet (UV) irradiation or involve complex chemical approaches to induce the bonding of polymers to substrates [17,18].

The technique that PNIPAAm and derivatives are grafted to the surfaces of polystyrene culture plates by EB irradiation is widely applied by far. This scheme was firstly developed and reported by Okano’s group for safe and mild separation of cell sheets from thermoresponsive culture plates grafted with PNIPAAm in 1990 [19,20]. Illustratively, the recrystallized NIPAAm monomers are first dissolved in 2-propanol and added evenly into culture plates, then irradiated by 0.25 MGy electron beam, and finally the grafted culture plates are repeatedly rinsed by distilled water and dried under vacuum [21]. Plasma polymerization is also a common method for fabrication of thermosensitive surfaces, which is available for preparing highly networked thin films without the help of crosslinkers [22]. The plasma deposition coatings have excellent physical properties. The films are sterile and compact, and entirely overlay and bind to the specified substrates. However, the monomers are vulnerable to rupture when subjected to plasma stimulus, so the active functional groups of monomers might be destroyed, and accordingly leading to the ineffective function of materials. Grafting polymerization by UV irradiation is able to change the surface chemistry (and thus surface properties) of polymer materials as well. Compared with other modification technologies, UV irradiation has its unique superiority in applications, like fast reaction speed, low energy consumption, simple procedure, easy industrialization as well as the reaction occurs only in shallow surface region of the substrate [23]. Owing to weak penetrativity of the ultraviolet ray, the treated substrates are seldom affected by irradiation, therefore, the surface photografting polymerization can only modify and adjust the surface properties and will almost never break the bulk materials. The thicknesses of grafting planar films are dependent on the reaction parameters such as irradiation time, reaction temperature, initiator concentration and so on.

Both atom transfer radical polymerization (ATRP) and reversible addition fragmentation chain transfer (RAFT) polymerization can provide special designing plans for preparation of thermosensitive surfaces [24,25,26]. ATRP polymerization often utilizes a transition metal complex like Cu, Fe, Ru and Ni as a catalyst with an alkyl halide as an initiator, while RAFT polymerization uses thiocarbonylthio compounds as chain transfer agents, such as dithioesters, thiocarbamates and xanthates. ATRP and RAFT polymerization can be applied under a wide range of reaction conditions, and they are available for compatibility with multiple functional monomers. However, it is hard to remove the impurities like transition metal ions and bipyridine from the resultant products in the ATRP process, while addition of thiocarbonylthio compounds might raise the risk of toxicity in the synthetic polymers during the RAFT polymerization, which lead to the difficulties for pretreatment of thermoresponsive planar films before usage as well as the adverse effects of cytotoxicity in the process of being used.

The above-mentioned surface modification methods are expectedly helpful to achieve in situ copolymerization of NIPAAm and other functional monomers, which are essential for covalent crosslinking of polymers as well as grafting between polymers and substrates, and to ensure the end products have good stability and durability and to make the film thickness approach nanoscale. However, since polymerization and grafting sites occur with random frequency, in situ copolymerization is inefficient to control the microstructure of polymer films. Nitroxide radical polymerization (NMP) is a remedy that makes use of alkoxyamine initiators to generate PNIPAAm mediated films with well-defined, functional and complex macromolecular architectures [27].

#### 2.1.2. Physically Adsorbed Coating Films

Unlike covalent grafting, it normally takes two successive steps to form physically adsorbed coating films, including preparation of PNIPAAm polymers and coating of polymer solutions [28,29]. Generally, PNIPAAm or derivatives are first synthesized and then sufficiently dissolved into a volatile solvent, and subsequently the polymer solution is coated onto a flat substrate and treated by natural or manual drying, and eventually the unstable coating is rinsed by deionized water or other solvents to remove non-adsorbed polymers, thereby forming a adsorption layer of thermosensitive polymers on substrate surface [28,30].

PNIPAAm homopolymers and copolymers can be prepared by bulk polymerization, emulsion polymerization, suspension polymerization, solution polymerization and so on. Solution polymerization is a simplest case. Benzoyl peroxide, peroxyacetic acid and azobisisobutyronitrile are often served as initiators for free radical polymerization. Water, alcohol, ether, acetone, tetrahydrofuran, chloroform and benzene are widely used as solvents. The properties of the synthetic polymers are typically subject to the great influences of the key technological parameters such as reaction temperature, reaction time, initiator dosage and monomer ratio [31]. PNIPAAm and derivatives can also be easily functionalized by means of chain transfer agents [32] or copolymerization of functional monomers with double bonds [33] via free radical polymerization. The former schemes are referred to as chain transfer reactions. The activity of a growing polymer chain is transferred to another molecule to form a derivative polymer with radical initiator end and thermosensitive group end, which allows for the polymers to be applied in various settings and applications. The latter routines involve the copolymerization of NIPAAm and other functional monomers to modify the multiple properties of the manufactured polymers to meet specific needs, so as to increase biocompatibility, change transition temperature, control surface wettability and alter mechanical property or improve biodegradability.

There are several techniques used to deposit PNIPAAm and derivatives as a thin film onto a substrate. The coating films with desired thickness and morphology can be easily tunable through the regulation of experimental parameters and technological variables. Drop-casting is an easier and more accessible method, which usually results in the spreading of polymer solution and the formation of thin film on relatively small substrates after solvent evaporation under programmed conditions [34]. Film thickness is dependent to solution concentration and dispersion volume. However, uncertain evaporation rates and concentration fluctuations can lead to the variations in film thickness or internal structure, namely poor uniformity. Compared with drop-casting, spin-coating is feasible and applicable to get more uniform films onto the given substrates and to accommodate with larger substrates [35]. A centrifugal force gives rise to uniform spreading of nanoparticle dispersion across the substrates, followed by solvent evaporation to obtain thin films. Film thickness can be controlled and repeatable, which is sensitive to solution properties, dispersion volume and rotational parameters. Dip coating is also a simple method of depositing PNIPAAm and derivatives onto the substrates, which is capable of producing very uniform, close-packed particle films [36]. This technique involves withdrawing a substrate at a certain speed from polymer solution and depositing nanoparticles on the substrate as the thin liquid layer dries. Film thickness is determined by the competition among viscous force, surface tension and gravity at the liquid–substrate interface. Spray-coating utilizes sprayers to generate homogenous and atomized droplets that settle evenly onto the targeted substrates [37]. This technique can use a variety of particle mixtures to a wide range of substrates and allows for tailoring of the resultant film deposition by manipulating the tunable parameters like solvent property, liquid flow rate, air pressure, sprayer geometry and position, etc.

The two-step preparation methods have some attractive traits. The coating process is relatively simple; the performance of coating films can be improved by adding functional ingredients during production process; the film thickness is tunable by selecting solvent, adjusting polymer concentration and changing coating conditions. However, the lack of crosslinking of intermolecular chains and further connections between polymers and substrates make the physically adsorbed films somewhat easier to detach from the substrates or dissolve in the working solutions when the ambient temperature is below LCST, namely poor stability and reversibility [38,39].

#### 2.1.3. Grafted Coating Films

The preparation strategies involving grafting or coating of PNIPAAm and derivatives onto the substrate surfaces are usually used for preparation of thermorespective planar films. However, it can be concluded from the mentioned descriptions that both approaches have their own weaknesses, such as applying expensive instruments and complicated techniques, introducing unconventional raw materials with potential toxicity, being difficult to remove impurities or reagents, insufficient stability, reversibility of end products, etc. This is not conducive to wide promotion and application of PNIPAAm mediated films, however, the predominance of grafting or coating cannot be ignored. If incorporating the both merits, it is bound to make a very feasible and powerful fabrication scheme for temperature-responsive films.

Our research group has authored PNIPAAm grafted films on silicon or glass based surfaces using a novel approach [33]. The two strategies are utilized successively during operation, so this method is named coating–grafting two-step film formation. Briefly, PNIPAAm copolymers with siloxane and hydroxyl groups are first synthesized, and subsequently the copolymer solution is evenly spin-coated or dip-coated on the uniform surfaces bearing residual hydroxyl groups (like glass and silicon). Silyl crosslinking generally begins during heating treatment. Condensation reaction between siloxane and hydroxyl groups leads to methanol removal, and newly formed bonds provide integration within the film and coupling with the substrate surface. The coating–grafting integrated method used in our studies offers a simple, economic and useful way to creating thermoresponsive surfaces compared with routine techniques.

### 2.2. Spatial Supports

The studies on thermoresponsive planar films based on PNIPAAm have made encouraging progress, however, plane techniques are intended to be used only in certain limited area with a certain theme, thus resulting in the inadequacy for scale-up to large quantities in industrial production. The three-dimensional (3D) systems enable the enormous expansion of target products to become feasible. Therefore, it has been gradually attempted to introduce PNIPAAm and analogues into hydrogels, microcarriers, scaffolds and other tridimensional systems, and consequently to produce thermoresponsive spatial platforms for supporting an extensive variety of applications, especially for cell culture applications.

#### 2.2.1. Thermoresponsive Hydrogels

Smart hydrogels derived from thermoresponsive polymers with crosslinked network are noteworthy at the ability to adjust sol–gel transitions in response to ambient temperature. The thermoresponsive hydrogels are hydrophilic and highly water-absorbing in aqueous solutions when the temperature is lower than LCST, whilst they undergo phase transitions from swelling to shrinking when heating up to above LCST.

The thermoresponsive hydrogels facilitate to imitate the structure, property and further microenvironment of native tissue, owing to their hydrated and interconnected porous structures. PNIPAAm mediated hydrogels are often prepared by precipitation polymerization, suspension polymerization, membrane emulsification and so on. Precipitation polymerization is a heterogeneous polymerization protocol that initiates as a homogeneous system in a continuous phase (usually water), where NIPAAm, initiator (commonly ammonium persulfate along with tetramethylethylenediamine) and crosslinking agent (generally *N*,*N*′-methylene-bisacrylamide, MBAAm) are fully soluble, but the formed crosslinked PNIPAAm becomes insoluble and thus precipitates upon initiation [40,41]. After precipitation, polymerization proceeds by absorption of NIPAAm and initiator into PNIPAAm particles. The particle size is often confined to micron and submicron levels. Suspension polymerization is also an uneven polymerization procedure that begins with a mixture of NIPAAm and crosslinker (usually MBAAm) in oil phase by means of mechanical agitation with the support of surfactant, and then follows by PNIPAAm polymerization to form polymer spheres [37]. The particle sizes are often distributed between 0.02 and 2 mm, which mainly depends on agitation speed, reactor type and stirrer style [42,43]. Membrane emulsification is another useful technique for producing crosslinked PNIPAAm gels. PNIPAAm aqueous solution is forced through the pores located at the microporous membrane into the oil-continuous phase [40]. The immiscible droplets are immediately generated and then get apart separately from the end of membrane pores with drop-by-drop. Controlled droplet sizes and narrow size distributions can be obtained, and the sizes of uniform droplets can be up to micrometers [44]. The droplets are sequentially transformed into solid particles by polymerization [45], solvent evaporation [46] and gelation [47].

#### 2.2.2. Thermoresponsive Carriers

The organic or inorganic carriers loaded with mono- or multi-responsive polymers are termed as smart carriers, which show excellent performance potentially usable in a wide range of fields particularly for cell culture. The use of thermoresponsive carriers facilitates intracellular and intercellular signaling, which plays vital roles in controlling the cellular functions and even cell fate, and allows for large-scale cell amplification and non-enzymatic cell harvesting in vitro [48].

In recent decades, PNIPAAm mediated carriers have been gradually developed by using different fabrication methods, and the technical key is establishing the successful linkage between polymers and substrates, as shown in Figure 3. The “grafting to” strategy is very popular and powerful, where the thermoresponsive carriers are typically formed by building the multipoint attachment between end-functionalized PNIPAAm and complementary reactive groups located at the surfaces of carriers [41], such as carboxyl-terminated PNIPAAm grafted to aminated glassbeads [49] and commercial Cytodex-3(R) microcarriers [50], and amino-ended PNIPAAm bonded with carboxylate groups on sodium alginate [51]. This method is experimentally simple and the resultant polymers can be identified before being grafted on the surfaces, but the steric hindrance caused by intermolecular polymer chains will lead to the limitation of grafting density.

The other common approach is known as “grafting from”. In this case, the molecular chains of PNIPAAm and derivatives continuously grow in situ from the reactive groups (typically polymerization initiators) that have been already immobilized covalently on carrier surfaces [52]. This strategy is conducive to the deposition of more plentiful molecular chains of polymers on the surfaces compared with “grafting to” method owing to having avoided steric hindrance, but it provides little control over the chain length and uniformity of the resulting polymers [53]. Many techniques including ATRP, RAFT and NMP that are assigned to surface-initiated controlled/living radical polymerization (CLRP) processes will contribute to overcoming the drawback [41].

Another alternative to fabricate thermoresponsive carriers is “grafting through” approach, which depends upon the copolymerization of the surface-tethered monomers with the growing PNIPAAm polymers initially formed in solution by virtue of bulk free radical polymerization. This scheme combines the elements of “grafting to” and “grafting from” approaches, and allows for the fine control of polydispersity, functionality, backbone length, branch length and reactivity ratio of monomers and PNIPAAm [54,55]. 

#### 2.2.3. Thermoresponsive Scaffolds

There are various forms of thermoresponsive scaffolds available for cell culture applications, such as porous matrix, fibrous mesh and so on. PNIPAAm-mediated scaffolds are typically formed by means of sphere-templating, freeze-drying, electrospinning and other suitable techniques. These techniques allow for the design and manufacture of PNIPAAm mediated scaffolds with highly porous and well-interconnected structure, as illustrated in Figure 4. These elaborate scaffolds with open texture facilitate cell–cell and cell–matrix interactions of therapeutic cells.

Sphere-templating scaffold fabrication has been greatly developed over the last decade. In this technique, the colloidal particles (like silica, gold, polymer latex, etc.) are first prepared by packing homogeneous spheres into template arrays, then the interstices are filled with a fluid which is subsequently transformed into solid framework through precipitation, polymerization and other efficient methods, and finally the sphere templates are removed by dissolving them into the selected solvent to form the interconnected solid skeleton [56]. The flexible options of sphere templates, template layout, interstitial filling and other synthesis conditions permit the formation of periodic porous solids with multiple pore sizes ranging from Angstroms level upwards, allow for the favorable control of material performances, and further make it possible in scaffold optimization for specific applications.

Freeze-drying technique is also chiefly applied for the fabrication of porous thermosensitive scaffolds, which is based upon the principle of sublimation. Briefly, a homogenous PNIPAAm mixture with preferred concentration is first introduced to molds, and then quickly frozen in liquid nitrogen to preserve the original structure, and eventually freeze-dried under vacuum to obtain highly porous scaffold with high interconnectivity [57]. This technique needs neither high temperature nor leaching treatment of porogen components, however, it often fails to get porous scaffolds with regular pores and requires the aid of appropriate surfactants to stable the emulsion. The pore size and porosity are dependent to the concentration and viscosity of polymer solution in continuous phase and the volume of aqueous phase dispersed in the system.

Electrospinning is widely employed in fabricating nonwoven thermoresponsive scaffolds, which is capable of producing polymer fibers in nanometer diameters. In this case, emitting and stretching of polymer solutions are easily triggered by use of high voltage and followed by solvent evaporation, leading to the formation of porous scaffolds with ultrafine fiber mat. It has good control over the pore geometry of nanofibers, like orientation, surface area and aspect ratio [57,58]. The design and manufacture procedure for electrospinning technique shows simplicity and high efficiency, which is developing toward the promotion of better cell activity.

## 3. Characterization of PNIPAAm Medicated Substrates for Cell Culture

An essential prerequisite for the successful applications of cell culture is careful assessment of comprehensive performances of PNIPAAm containing substrates by various analytical techniques. It is considerable to determine the specified features of temperature-responsive substrates, including chemical composition, molecular weight, spatial scale, thermosensitive character, swelling property, mechanical property, grafting density, film thickness, surface topography, surface wettability, etc. The characteristics of thermoresponsive substrates strongly influence cell attachment behaviors and detachment properties, thereby profoundly determining life spans of targeted cells, so it is necessary to take all of these features into consideration during the selection or design work [59]. The commonly used physical and chemical analysis techniques for PNIPAAm mediated products are summarized in this review, as displayed in Table 1. Many other robust techniques are equally employed, but they will not be covered herein.

### 3.1. Identification of PNIPAAm Homopolymers and Copolymers

PNIPAAm homopolymers and copolymers cannot be used directly in practical applications, however, they are the fundamentals for the subsequent production of functional substrates. Therefore, the properties of thermoresponsive polymers are the key factors for film preparation and performance adjusting.

#### 3.1.1. Chemical Composition

The existing elements and their contents in PNIPAAm mediated materials can be determined by element analysis [60], which generally analyses carbon, hydrogen, oxygen, nitrogen, sulfur and other elements, and particularly detects the relative contents of carbon, hydrogen and oxygen elements. The ratio of carbon, hydrogen and oxygen in PNIPAAm homopolymers is commonly fixed, whereas there are more complexities about the additions of ingredients in PNIPAAm copolymers, thus leading to the uncertainty of the element ratios. The molecular proportion of the reactants bearing in thermoresponsive copolymers can be roughly inferred from the results of element ratios, but it also needs to be complementally verified by other assessment experiments.

The characteristic absorption peaks appearing in the spectra of Fourier transform infrared spectroscopy (FTIR) can be used to illustrate the functional groups of PNIPAAm containing samples. The ratio of each component in a hybrid complex can also be deduced from the peak intensity. The stretching vibration absorption peak of amide bond I and the bending vibration absorption peak of amide bond II observed at 1650 and 1550 cm^−1^ respectively, and the symmetrical deformation vibration absorption peaks of isopropyl group located at 1387 and 1367 cm^−1^, are representative absorption peaks of PNIPAAm [61]. The definite molecular structure of PNIPAAm based compounds can be obtained by interpreting the information of chemical shift, peak shape, integral area and coupling constant derived from nuclear magnetic resonance (NMR) spectroscopy [62]. It is clearly discovered from NMR spectra that a strong peak presents at around 1.15 ppm, which belongs to the isopropyl methyl protons (–CH_3_) of PNIPAAm chains [63].

#### 3.1.2. Molecular Weight

Size-exclusion chromatography (SEC) is normally employed to determine the molecular weight and molecular weight distribution of PNIPAAm and derivatives [64,65]. Static light scattering (SLS) enables to measure the average molecular weight of macromolecules in PNIPAAm solution [66], but this technique requires the system to be sensitive and exceptionally stable. SEC testing has to separate the components of a sample before the calculation of an accurate molecular weight distribution, whilst an estimate of the molecular weight can be made in the SLS technique, which has the benefit of being faster to confirm the oligomeric state than the SEC measurement. Mass spectrometry (MS) is an analysis technique that ionizes chemical species and ion sorts on the basis of mass-to-charge ratio under high vacuum [67], and it can be applied to measure the molecular structure and molecular weight of thermosensitive polymers as well.

#### 3.1.3. Volume Phase Transition Temperature/Lower Critical Solution Temperature

The critical transition temperature for the sharp changes of hydrodynamic diameters is defined as volume phase transition temperature (VPTT) of thermoresponsive microgels (or nanogels), analogue to the LCST for PNIPAAm based polymers [68]. Differential scanning calorimetry (DSC) is a thermal analysis technique. In this case, the difference in the energy required to elevate the temperature between a specimen and a reference is evaluated as a function of temperature. It specializes in exploring the phase transition temperature, crystallization temperature, structural relaxation peak as well as crystallization enthalpy change of PNIPAAm mediated materials [69].

Dynamic light scattering (DLS) can be used to determine the LCST of thermoresponsive PNIPAAm and derivatives, based on the prediction of interaction behaviors between polymers in the system. Across LCST, the particle sizes of PNIPAAm containing products display an abrupt change from large to small or show a sharp transition from small to large [70,71,72]. The LCST of PNIPAAm homopolymers and copolymers can also be simply measured by cloud point (turbidity) measurements. The optical transmittances of the solutions are monitored at 500 nm wavelength via UV-Vis spectrometer with incremental temperature interval. The inflection point of turbidity plots is regarded as the LCST of the synthetic polymers [65].

### 3.2. Characterization of Thermoresponsive Planar Films

#### 3.2.1. Chemical Composition

X-ray photoelectron spectroscopy (XPS) is an important surface-sensitive quantitative spectroscopic technique, which is available to analyze the elemental and chemical composition of the polymer coatings. XPS spectra are generally obtained by irradiating dried materials with X-rays while monitoring the number of electrons escaping from the topmost 10 nm of the material surfaces [16]. Therefore, XPS technique can provide the explanatory notes related to the chemical composition and coating uniformity of PNIPAAm based surfaces. High-resolution XPS can also offer the helpful message on chemical structure of thermoresponsive layers based on the chemical shifts and states of the functional groups from PNIPAAm and derivatives [73]. Sum frequency generation (SFG) spectroscopy is non-destructive in-situ testing method, and it is dispatched to the measurements of chemical composition, orientation distribution and structural information at the outermost surfaces or two-phase interfaces. This technique is sensitive to the monolayer surfaces, and it has little damage to material surfaces [74]. Therefore, the chemical information of PNIPAAm mediated substrates is held in ease by using the SFG technique. Time-of-flight secondary ion mass spectrometry (TOF-SIMS) is also an ideal surface inspection technique with high sensitivity and specificity, and it can give more detailed information about the molecular composition of the topmost layer of thermoresponsive coating or films [25,75,76].

#### 3.2.2. Grafting Amount

Grafting amount of PNIPAAm and derivatives onto a solid substrate is a considerable factor affecting cell attachment and detachment [77]. The grafting amount of polymers is easily determined by the gravimetric method and calculated on the basis of the mass difference before and after the respective functionalization per outer surface area. This simple method is only suitable for the measurement of moderate grafting amount, but it is incapable with lower grafting amount of thermoresponsive polymers. Attenuated total reflection (ATR) is a responsible and attractive technique usually associated with FTIR (namely ATR-FTIR), which is appropriate for direct determination of the characteristic functional groups of the specimens in either solid or liquid state without further treatment. Hence, the grafting amount of thermoresponsive grafted films can be investigated by means of ATR-FTIR technique [77]. It is clearly illustrated with an example of PNIPAAm grafted tissue culture polystyrenes (TCPS). For TCPS, the characteristic absorption peak appearing at 1600 cm^−1^ is allocated to monosubstituted aromatic rings, while for PNIPAAm containing layers, the clear absorption peak of amide bond I can be found at 1650 cm^−1^. The grafting amount of PNIPAAm on benzene based surfaces can be deduced by calculating the peak intensity ratio of I_1650_ to I_1600_ from ATR-FTIR spectra [20,78,79].

#### 3.2.3. Surface Topography

Physical features of thermoresponsive surfaces especially topography have proven to have remarkable impacts on cell behaviors [59]. Thus it is very necessary to choose appropriate techniques to investigate the surface morphology for PNIPAAm mediated materials. Scanning electron microscopy (SEM) gives high-resolution images of specimen surfaces through a centralized electron beam that is scanned over the specimens under vacuum, and accordingly provides the topographical signals of specimens and further information about chemical composition of samples. SEM can reveal the image details of a sample with best resolution of 1 nm under right circumstances. Atomic force microscopy (AFM) is a very important nanoscale microscopy with higher resolution than SEM. The information is collected by feeling or touching sample surfaces with a mechanical probe. Both SEM and AFM are frequently used to analyze the microstructure of thermoresponsive surfaces [80,81], and further to investigate the changes of surface topography with respect to temperature variation. The surface topography is generally displayed as 3D profiles with pseudocolor plots in AFM technique, whereas it is showed as 2D projections with variations in brightness in SEM images. In addition, PNIPAAm mediated materials are commonly covered with an ultrathin coating of electrically conducting materials (such as gold and carbon) to increase their electrical conductivity and to stabilize them before testing, so that they can withstand high vacuum and high electron beam. Thermoresponsive samples are easy to be observed upon AFM without special treatment, and they are usually not affected by the charging artifacts in the final images. Profilometer is also a useful tool in quantifying the surface roughness, and accordingly shows the related critical dimensions like step, curvature, flatness of thermoresponsive planar films [82].

#### 3.2.4. Surface Wettability

Surface wettability is of extreme importance in protein adsorption or desorption [18,67,71,83]. Such a factor will no doubt duly impact cell attachment and detachment behaviors. Surface wettability is often quantitatively evaluated using static contact angle. This quantitative index is often measured by sessile drop method via optical arrangement, and determined through a goniometer at given temperature and pressure. Contact angle is more susceptible to surface roughness, surface heterogeneity and swelling behavior [84]. At the temperature below LCST, lower contact angles reveal more hydrated surfaces of PNIPAAm mediated films. The rising temperature to above LCST pushes the contact angles even higher, indicating that the thermoresponsive films are water-repellent in this case. Thus contact angle analysis is competent in explaining the thermosensitivity of PNIPAAm containing layers.

#### 3.2.5. Film Thickness

The thickness changes and wettability variation of PNIPAAm based films are considered as the main driving forces causing effective cell attachment and spontaneous cell detachment, so the assessment of film thickness is necessary before cell culture. Spectroscopic ellipsometry (SE) is an effective measurement technique, and it has extraordinary abilities to characterize the metrology of thin PNIPAAm grafted or coated films, including composition, roughness, thickness, electrical conductivity and other properties [18,24,33]. However, conventional SE technique works only for the measurement for dry samples. Specially designed devices are required to determine the changes of film thickness with temperature variation in aqueous solutions [33,85]. Surface plasmon resonance (SPR) can measure the wet thickness of thermoresponsive layer in liquid environments in view of swelling behavior [64]. It is also difficult to estimate the thickness of PNIPAAm grafted on polymeric substrates by optical SE and SPR, since PNIPAAm and substrates have similar refractive index. The combination of UV excimer ablation technique and AFM measurement can be useful to determine the thickness of PNIPAAm on TCPS surfaces [20,78].

### 3.3. Characterization of Thermoresponsive Hydrogels

The chemical features for thermoresponsive hydrogels can still be measured by conventional means like FTIR, NMR and XPS [86,87,88], but there are several physical characteristics that require special measurements.

#### 3.3.1. Thermoresponsive Swelling

Thermoresponsive swelling is one of the most interesting features for PNIPAAm based hydrogels. DLS is a usable technique for the evaluation of nanogel swelling, but it is unserviceable to gels with larger size [89]. DSC is capable of detecting the swelling behaviors of thermoresponsive hydrogels as well [90]. ATR-FTIR technique can also be used for the analysis of the effects of bonding formation and decomposition on the swelling behaviors [84]. Dynamic rheology is a routine approach to examining the rheological behaviors of PNIPAAm mediated hydrogels, which can be employed to evaluate sol–gel transitions with temperature shift. The rheological properties of homogeneous PNIPAAm microgels are dependent to ambient temperature, shear rate and solution concentration [91]. Naturally, the most straightforward strategy is the gravimetric method. Typically, dry thermoresponsive hydrogels are immersed into purified water or saline buffers, and the hydration keeps stable against the entire thermal range. The mass or dimension (like thickness, height and diameter) difference before and after absorbing water against time is viewed as the kinetics of swelling [86,87].

#### 3.3.2. Mechanical Properties

The evaluation of mechanical properties can be used to determine the durability and further the integrity of PNIPAAm based hydrogels. A simple method is to submit the samples to a specific shear-flow. Compression assays by means of a rheometer can also give the useful information about the mechanical properties of thermoresponsive gels. The mechanical stress can serve as the detection means as well. In this case, PNIPAAm mediated hydrogels are first treated by stirring, and the broken and undamaged hydrogel is quantified by visual analysis with the aid of stereomicroscope. Mechanical properties can be further evaluated by swelling and stability measurements [92].

#### 3.3.3. Architecture

Raman microscopy is an efficient and nondestructive way to image the pore structure of PNIPAAm based macrogels, but lower resolutions make it impossible to use for the characterization of microgels (or nanogels) [89]. SEM is often used to monitor the peripheral signs and internal structure of thermoresponsive hydrogels, but the pretreatment of materials is required. Briefly, the samples for SEM images are firstly soaked in water and stand to full swelling at required temperatures ranging from above LCST to below LCST, then placed into liquid nitrogen immediately for a short time to “freeze” the hydrogel structures, and eventually dehydrated with a freeze drier for about two days to remove water content entirely [9]. Temperature sensitivity is reflected by aperture changes of lyophilized hydrogels. The thermoresponsive hydrogels undergo constant adjustment of conformation when after being processed by cooling treatment, and the pore sizes of gels usually increase several times as compared to the original scale at higher temperature.

### 3.4. Characterization of Thermoresponsive Carriers

#### 3.4.1. Chemical Features

The chemical components grafted on the carrier surfaces are routinely characterized by elemental analysis, ATR-FTIR and NMR techniques [93,94,95,96,97,98,99]. There are also some powerful tools available for determining the chemical components of thermoresponsive carriers. As described, XPS can provide informative data concerning the chemical features of the outermost surface within several nanometers, thus it can serve in the evaluation for the chemical composition of thermoresponsive carriers. A new N1s signal with binding energy at 400 eV appears in XPS spectra after the grafting of PNIPAAm, indicating that PNIPAAm are bound to the carriers successfully [99]. The element ratio of carbon, nitrogen and oxygen can also be determined from XPS analysis, which indirectly reveals the grafting amount of PNIPAAm. Energy dispersive spectroscopy (EDS) technique can be used for qualitative and semi-quantitative analysis of PNIPAAm mediated materials. EDS is typically coupled with SEM, which allows for the chemical analysis of sample features being observed in SEM monitor. This combined SEM/EDS system can provide morphological images as well as chemical information [100]. The appearance of a nitrogen peak in EDS profile demonstrates the successful conjunction between PNIPAAm and carriers [93,98,99]. The grafting amount of PNIPAAm onto the carrier surfaces can be deduced from the nitrogen content increase obtained from electron spectroscopy for chemical analysis (ESCA, known as XPS) [91,93].

#### 3.4.2. Size and Morphology

The size and morphology of thermoresponsive carriers can be favorably monitored using microscopy instrumentations, such as optical microscopy, SEM, AFM and so on [93,94,95,96,97,98]. Therein, bright field and fluorescence microscopy is most popular, because this technique is able to rapidly and effectively give preliminary information about preparation of PNIPAAm modified carriers before more complicated techniques are exploited [92].

#### 3.4.3. Thermoresponsive Swelling

For core materials at the nanoscale, the thermosensitive characters of PNIPAAm grafted carriers can be assessed by rheological experiments. The viscosity of thermoresponsive nanocarriers will be amplified to thousands of times at the temperatures changing from below LCST to above LCST at low shear rate [94]. DLS can also be used to determine the swelling ratio of thermoresponsive carriers with the hydrodynamic diameters of the swollen particles less than 1 μm under desirable temperatures [93,94,101]. For micro-sized and macro-scale particles, the thermosensitive swelling of hybrid materials can be determined by optical observations. In this case, the average diameters of PNIPAAm based microcarriers are obtained by measuring several hundreds of particles from the pictures taken by digital camera that is connected with the microscope [95,97].

### 3.5. Characterization of Thermoresponsive Scaffolds

#### 3.5.1. Chemical Features

Many techniques used for the evaluation of thermoresponsive substrates are also capable of determining the chemical structures of thermoresponsive scaffolds, which are frequently characterized using ATR-FTIR, NMR, XPS and SEM/EDS systems [65,102,103,104,105]. Grafting yield of PNIPAAm mediated scaffolds can be assessed by analyzing the ratio of peak intensities of characteristic and reference peaks from ATR-FTIR spectra. The grafting degree is also evaluated by XPS analysis according to the actual nitrogen concentration of the grafted surfaces compared with the theoretical nitrogen content in PNIPAAm polymers [104].

#### 3.5.2. Morphology and Structure

Scaffold microstructure (surface and cross-section) is commonly evaluated by SEM, AFM and light microscope like thermoresponsive macrogels [102,103,104,106,107,108]. Pore diameter and distribution or fiber size and alignment are often analyzed by randomly picking at least dozens of various segments by means of image J software [65,109]. Individual scaffold dimensions can be measured with digital calipers [108,110]. Scaffold porosity can be determined using liquid displacement method [104]. The criteria for excellent displacement liquid is that it can easily penetrate into the scaffold pores and does not cause a change in volume, and the solvent will not dissolve the prepared scaffolds. The density quantities give useful information concerning aperture, dispersity, penetrability and structural deficiency of scaffolds [38,65].

The nitrogen multilayer adsorption method can be used to obtain the accurate specific surface evaluation of PNIPAAm based materials with relative pressure as experimental function via an automatic analyzer, upon Brunauer–Emmett–Teller (BET) surface area analysis. This technique identifies the total specific surface area by evaluating the external and pore areas, thus providing helpful message for studying the impacts of surface porosity and particle size in a variety of applications. Barrett–Joyner–Halenda (BJH) measurement can be used to estimate pore area and specific pore volume via nitrogen adsorption and desorption techniques as well. BJH technique characterizes pore size distribution irrelevant to external area owing to particle size of the specimen [106].

#### 3.5.3. Surface Wettability

It is generally accepted that contact angle measurements are commonly applied to analyze the surface wettability and surface energy of materials, but they are generally not useful for porous surfaces owing to the heterogeneity, capillary force in pores and surface reconstruction [111]. However, comparison between poriferous samples with temperature changes can still be effectively made. The contact angles confirm that the thermoresponsive nonwoven fabric is thermoresponsive with a relatively water-repellent surface at 40 °C with enhanced hydrophilicity at 20 °C [104].

#### 3.5.4. Thermoresponsive Swelling

The swelling ratio is relevant to the hydrophilicity/hydrophobicity of thermoresponsive scaffolds. The gravimetric method is a well-trodden technique and also deployed extensively. The water absorption rate of poriferous scaffolds is determined by soaking them in water or buffer solutions at a required temperature for a predetermined time. The thermoresponsive swelling can be obtained by analyzing the mass difference of dried and hydrated scaffolds [65,68,102,106].

#### 3.5.5. Mechanical Properties

The mechanical characteristics of thermoresponsive scaffolds can be evaluated by compression testing instrument [65,108]. The stress–strain data like ultimate tensile strength, strain at break and tensile modulus is measured by load-displacement analysis and the compression modulus is computed from the elastic region of the stress–strain plot [65,107].

### 3.6. Biocompatibility Assessment of Thermoresponsive Substrates

#### 3.6.1. Cell Viability

The evaluation related to the biocompatibility of biomedical materials is an indispensable step before clinical experiment, which is usually characterized by cell viability. Several approaches are available for such determination for PNIPAAm modified substrates, but indirect methods using fluorescent or colorimetric indicators can offer rapid and large-scale assays.

Trypan blue staining is routinely applied to assess cell viability via dye exclusion test, which is often performed while counting cells with hemocytometer [112,113,114]. Trypan blue, an impermeable dye, is able to enter dead cells with damaged membranes, but this dye cannot penetrate into living cells. Calcein-acetoxymethyl ester (calcein-AM)/ethidium homodimer staining is another common method to measure the viability of therapeutic cells. The permeable calcein-AM is enzymatically converted into calcein showing bright green fluorescence in the cytoplasm of viable cells, while ethidium homodimer can enter cells with compromised membranes and takes up red fluorescence upon binding to nucleic acids [114]. The ratio of fluorescence intensity of calcein AM to ethidium homodimer demonstrated the degree of cell viability [5,115].

3-(4,5-dimethylthiazol-2-yl)-2,5-diphenyltetrazolium bromide (MTT) is a standard and popular colorimetric assay for measuring cellular activity, and cell counting kit 8 (CCK8) also provides a simple and robust way to conduct cell viability assessment [35,116,117]. MTT assay involves the conversion of water-soluble MTT to insoluble formazan, which is catalyzed by mitochondrial succinate dehydrogenase derived from live cells. The concentration of formazan depends on mitochondrial respiration, and it is easily measured by optical density at 570 nm using standard absorbance readers. CCK8 analysis utilizes water-soluble tetrazolium salt to quantify the number of viable cells by generating orange soluble formazan dye upon bioreduction of cellular dehydrogenases. The amount of formazan produced is directly proportional to the number of living cells and is determined by absorbance at 460 nm.

#### 3.6.2. Cell Attachment and Detachment

The PNIPAAm modified substrates exhibit an interesting temperature-responsive behavior with temperature variation, so the cells can adhere and grow readily on hydrophobic surfaces at physiological temperature, whilst the cells detach easily from hydrophilic substrates when cooling down to below LCST [118]. The evaluation of cell attachment on PNIPAAm modified substrates, as compared with pristine substrates, is often conducted at 37 °C for less than 24 h; the assessment of cell detachment from thermoresponsive surfaces is performed upon the temperature ranging from 10 to 25 °C for some time after routine culture; both of the procedures are followed by cell washing. The fraction of cells which remain adhered or separate from culture substrates after washing reflects the attachment/detachment rate of target cells, which can be generally determined by direct cell counting, spectrophotometric measurement (like MTT and CCK8 assay) and DNA quantification [112,113,116,117,119,120]. In addition, cell morphology and number can be examined by optical microscope, fluorescent staining and SEM imaging, which indirectly explicate the adhesion/deadhesion level of the cells upon a change in temperature [5,35,115,116,117,121].

## 4. PNIPAAm Medicated Substrates Used for Cell Culture

PNIPAAm is a valuable thermosensitive polymer, which is extensively used in biomedicine fields. Here we highlighted the important applications of PNIPAAm for cell culture. The smart PNIPAAm and matching thermal-liftoff method is an effective alternative to enzymatic and mechanical pathways. There is growing evidence that this unique cell culture platform not only offers suitable growth conditions for adherent cells under physiological environments, but also provides efficient and mild cell recovery by means of lowering temperature. As demonstrated in Figure 5, the targeted cells can be seeded on the thermoresponsive surfaces or embedded into the PNIPAAm based substrates when applied. For the former case, the cells readily adhere and grow on the thermoresponsive substrates with weak hydrophobicity and structural shrinkage above LCST, whilst the attached cells detach spontaneously from the hydrophilic surfaces owing to the hydrated and extended PNIPAAm molecular chains below LCST [122]. In the latter case, the cells are elaborately encapsulated in the PNIPAAm mediated hydrogels or scaffolds under room temperature and trapped at elevated temperature; cooling treatment increases the pore sizes of crosslinked gels/scaffolds or dissociates noncrosslinked supports, and consequently causes cell removal. The PNIPAAm mediated intelligent cell culture system tactfully controls cell attachment and detachment, which mainly takes sufficient advantage of surface wettability alterations and/or bulk structure variations with temperature changes. Thermally induced cell detachment allows for targeted cells to maintain stable growth, rapid proliferation and strong differentiation, since the adherent cells separate from temperature-responsive substrates with integral membrane proteins and most intercellular junction proteins [12,13].

The studies on 2D thermoresponsive substrates have achieved considerable achievements in cell culture and recovery. Further attempts have been made to find better ways to facilitate disposed adhesion and rapid detachment of therapeutic cells on the thermoresponsive surfaces, such as adjusting feed ratio [123], introducing hydrophilic groups [124], manipulating film thickness [125], changing surface roughness [126], creating patterned surface [127], setting optimum preparation condition [128], determining optimized storage environment [129], selecting special basal material [130], providing appropriate driving force [131], etc. Recently, stereoscopic culture platforms have witnessed rapid development and swift improvement of thermoresponsive microcarriers [132], scaffolds [103], hydrogels [133], hollow fiber membranes [134] and other 3D systems. Although the establishment and characterization of tridimensional culture systems based on PNIPAAm and analogues are undoubtedly more difficult and sophisticated than those of 2D systems, 3D systems have more applying value and developing foreground. The advanced culture models can maintain the structural and material basis of microenvironment in vivo, demonstrate the transparency and controllability during in-vitro cell culture, promote the collaboration of cell culture techniques and tissue engineering applications, therefore they can be employed in large-scale and nondestructive expansion of adherent cells.

## 5. Conclusions and Outlook

PNIPAAm is a powerful smart polymer that has high plasticity, making it versatile in biomedical fields. The studies of cell culture analogue systems involved thermoresponsive PNIPAAm and analogues have made great progress. The surface wettability and/or volume (phase) changes of PNIPAAm mediated substrates in response to temperature shift are the decisive factors for effective cell adhesion and spontaneous desorption. Researchers have been working to exploit advanced preparation strategies and seek appropriate characterization techniques for thermoresponsive substrates to facilitate willing attachment and fast detachment of targeted cells. This paper is dedicated to provide a comprehensive review of published articles addressing the preparation and characterization of PNIPAAm based substrates for cell culture applications. It covers the preparation schemes of thermoresponsive planar films and spatial supports, and also includes the characterization techniques specialized for different thermoresponsive substrates.

Many techniques capable of producing PNIPAAm based substrates have been reported but still require further investigation to make commercial manufacture possible. The fundamentals in the related areas such as dominant mechanism and critical influences on cell behaviors need to be further developed. Novel PNIPAAm based substrates with specific morphology and structure will be inevitably designed and prepared by use of new protocols in the future. The control over morphology, structures and further properties of thermoresponsive culture platforms has arguably become a recent focus of the study, and is likely to be future major development direction. Efforts to well improve the reproducibility, reliability and safety of PNIPAAm mediated materials in cell culture applications are also highly encouraged. 

The smart culture substrates are gradually transforming from monofunctional, planar films with single production to multifunctional, tridimensional supports with batched production, but remain to be further studied and developed. Here are some additional suggestions for establishing effective and well-organized intelligent stimulus-responsive platforms for cell culture: associating with excellent industrial production models, such as fluidized bed, 3D printing and microfluidic techniques to develop green, simple and economical fabrication schemes to promote the popularization of PNIPAAm and analogues in cell culture applications; screening comprehensive and reasonable physical and chemical characterization techniques to enhance the safety, reliability and feasibility of PNIPAAm based substrates, and further establishing complete and acknowledged identification programs for thermoresponsive culture substrates; investigating various external and internal parameters affecting cell behaviors and finding the optimal combination of existing influence factors to promote cell growth and harvest; adding nonhazardous functional compounds to upgrade the comprehensive properties of temperature-responsive culture platforms, like responsiveness, reusability, as well as biocompatibility and even to get valuable multi-responsive systems with virtues from multiple moieties; and combining PNIPAAm modified materials with dynamic culture techniques to expand ideal seed cells with fine quality in large quantities.

Although it is somewhat difficult to accurately predict a suitable trend for temperature-responsive systems in the coming days, it is hopefully that this systematic and focused review will be an inspiration for scholars to further proceed and intensify studies on PNIPAAm containing materials.

## Figures and Tables

**Figure 1 polymers-12-00389-f001:**
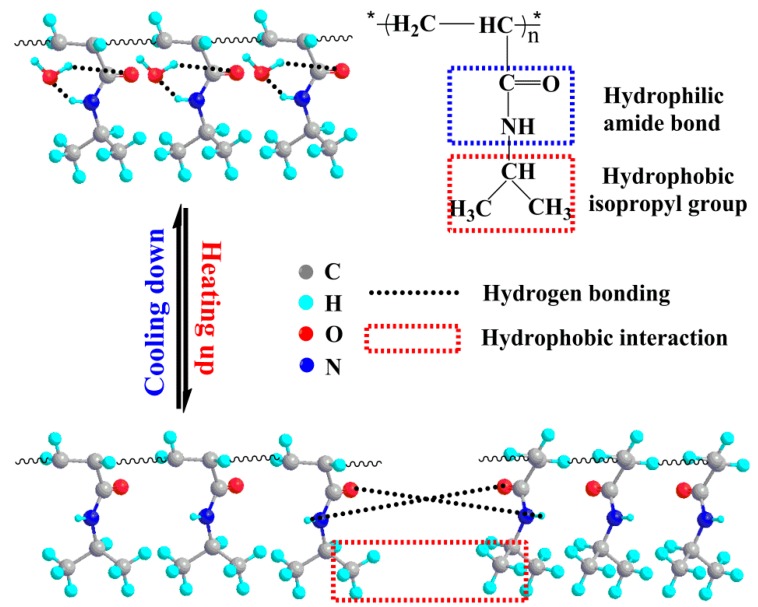
Molecular structure and thermoresponsive mechanism of PNIPAAm.

**Figure 2 polymers-12-00389-f002:**
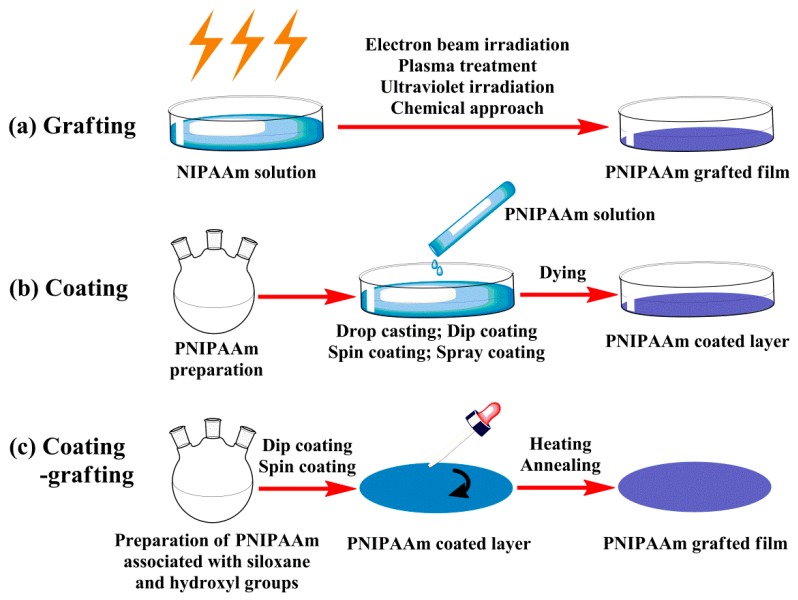
Preparation schemes of thermoresponsive planar films: (**a**) grafting, (**b**) coating and (**c**) coating–grafting.

**Figure 3 polymers-12-00389-f003:**
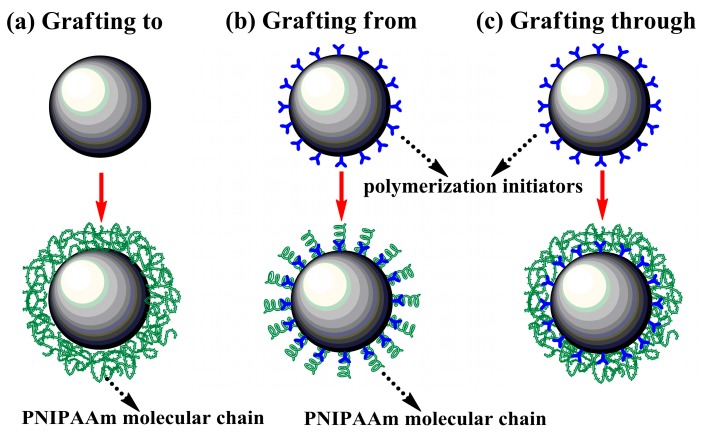
Fabrication strategies of thermoresponsive carriers: (**a**) grafting to, (**b**) grafting from and (**c**) grafting through.

**Figure 4 polymers-12-00389-f004:**
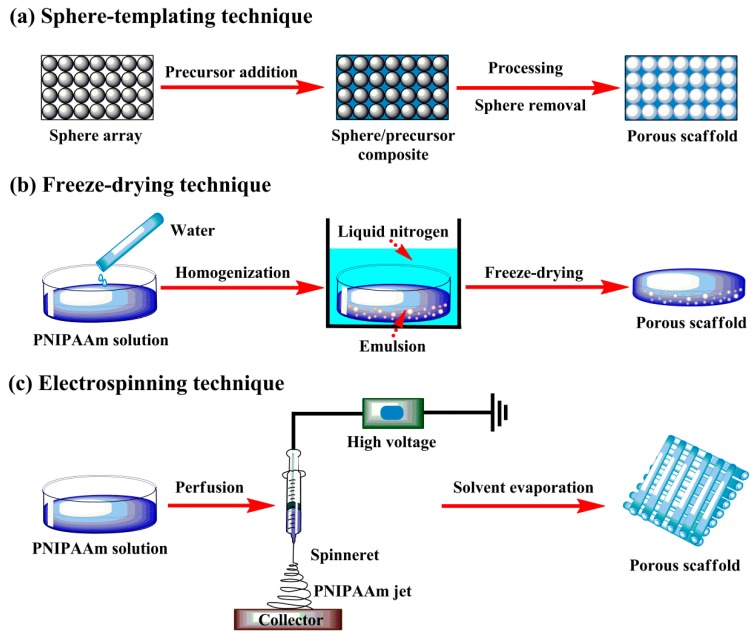
Manufacture techniques for thermoresponsive scaffolds: (**a**) sphere-templating, (**b**) freeze-drying and (**c**) electrospinning.

**Figure 5 polymers-12-00389-f005:**
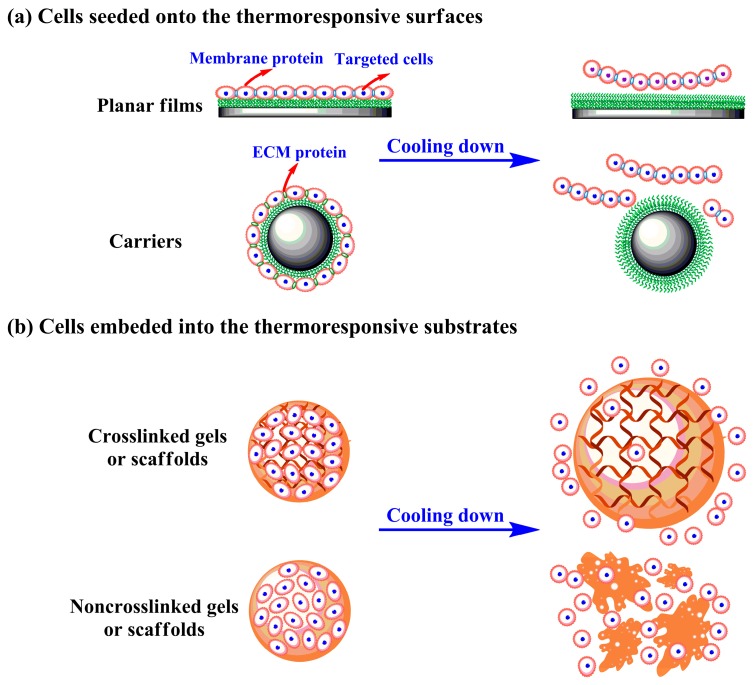
The targeted cells are seeded on the thermoresponsive surfaces (**a**) or embedded into the PNIPAAm based supports (**b**) above lower critical solution temperature (LCST), and harvested by cooling treatment.

**Table 1 polymers-12-00389-t001:** Characterization of thermoresponsive substrates for cell culture.

Substrate Type	Measurement Data	Detection Techniques
PNIPAAm homopolymers and copolymers	Chemical composition	Element analysis, FTIR, NMR
	Molecular weight	SEC, SLS, MS
	Phase transition temperature Lower critical solution temperature	DSC, DLS, turbidity measurement
Thermoresponsive planar films	Chemical composition	XPS, SFG, TOF-SIMS
	Grafting density	Gravimetric method, ATR-FTIR
	Surface topology	SEM, AFM, profilometer
	Surface wettability	Contact angle measurement
	Film thickness	SE, SPR
Thermoresponsive hydrogels	Chemical composition	FTIR, NMR, XPS
	Thermoresponsive swelling	DLS, DSC, ATR-FTIR, dynamic rheology, gravimetric method
	Mechanical properties	Shear-flow treatment, compression assay, mechanical stress
	Architecture	Raman microscopy, SEM
Thermoresponsive carriers	Chemical composition	Elemental analysis, ATR-FTIR, NMR, XPS, SEM/EDS system
	Size and morphology	Optical microscopy, SEM, AFM
	Thermoresponsive swelling	Dynamic rheology, DLS, optical microscopy
Thermoresponsive scaffolds	Chemical composition	ATR-FTIR, NMR, XPS, SEM/EDS system
	Morphology and structure	SEM, AFM, optical microscopy, digital calipers, liquid displacement method, BET, BJH
	Surface wettability	Contact angle measurement
	Thermoresponsive swelling	Gravimetric method
	Mechanical properties	Compression assay, mechanical stress

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
