# Peer review of "Preparation and Characterization of Thermoresponsive Poly(*N*-Isopropylacrylamide) for Cell Culture Applications"

_polymers, 2020, doi:10.3390/polym12020389_

Round 1
Reviewer 1 Report
The work entitled "Preparation and characterization of thermoresponsive poly(N-isopropylacrylamide) for cell culture applications" provides a concise, clear and well structured overview of the preparation and characterization of thermoresponsive substrates, namely PNIPAAm, for cell culture applications. The authors did a great job organizing the manuscript and selecting the main points for focus. It provides an interesting reading that is in no way tired and complex for the reader. I would only recommend the authors explore further their suggestions and future perspectives in the conclusion section and that in Table 1 they alter the positioning of the information in the first column since it is very difficult to read at the moment. Overall, the manuscript is well written with very little English mistakes (completely comprehensible considering the size of the manuscript), and the information is of significance. I recommend its publication after minor revision.
Author Response
Dear Reviewer, many thanks for your high praise and professional suggestions for this paper. We have revised Table 1 (Revised paper, Page 14) in accordance with your excellent advises. In the section of Conclusion and outlook (Revised paper, Page 16, Line 606-646), we have provided the information about current deficiencies, relevant suggestions and future perspectives of the preparation and characterization of thermoresponsive PNIPAAm for cell culture applications, and have added the description about content and significance of this review as well. We also have polished this manuscript and now submit to you for further consideration of publication. Please see more details in our revision. We hope the revised manuscript can satisfy you and meet the high standard requirement for the esteemed Journal. Thank you very much for your attention and consideration.

Reviewer 2 Report
The revision paper entitled as “Preparation and characterization of thermoresponsive poly(N-isopropylacrylamide) for cell culture applications” appears as extremely interesting not only for the hot topic devoted to, but to the very complete browsing through the actual state of the preparation and characterization of PNIPAAm .
The authors perform a very interesting introduction section, followed by a complete viewpoint of the preparation of the polymer for cell cultures. In fact, the article classified the preparation procedures to obtain Planar Films and Spatial Supports.
Additionally, they describe that Planar Films can be obtained by Covalent Grafting by EB, Plasma, UV irradiation, and by ATRP and RAFT procedures. In this sense, the description of the existing methods is exhaustive.
Equally, they also describe that these types of films can be prepared by physically absorbed coating by dip coating, drop casting, spray coating and spin coating. And finally, a hybrid method called coating-grafting.
The other support for cell cultures, the spatial supports procedures are also very well described by distinguish between thermoresponsive hydrogels, the three different types of thermoresponsive carriers, and finally, the many options to obtain the support by thermoresponsive scaffolds by sphere-templating, freeze-drying and elestrospinning techniques.
In a second part of the article the authors perform a very good work in describing the literature works in identifying the PNIPAAm homo- and co-polymers by the determination of the chemical composition (FTIR, NMR, element analysis, XPS, SFG, TOF-SIMS), the molecular weigth, VPTT and LCST, the grafting density (ATR-FTIR), surface topography (SEM and AFM), Surface wettability (contact angle), film thickness (SPR, SE).
Additionally, the characterization of the hydrogels is also exhaustive explored though the swelling (DSC, ATR-FTIR, Rheology, gravimetry), mechanical properties and architecture.
Similarly is done for the characterization of the carriers and scaffolds, and all these concerns and all the before mentioned has been compiled in a table for a better knowledge by the readers.
Finally, the article finish with PNIPAAm substrates used for cell culture and how the cells are or seeded onto or embedded into the thermoresponsive substrates.
The reading of the article provides a very complete panorama of the topic. Besides, the revision article provides 120 citations since 1994 up to 2019, with a great amount of recent cites.
This reviewer has enjoyed very much from the reading of the article, and would like to congratulate the authors from the great and interesting work performed.
Consequently, this reviewer would like to recommend the publication of the Revision Article in its actual state.
Author Response
Dear Reviewer, we would like to thank you for your careful reading, elaborate comments and high compliment. We have revised and polished the manuscript and now submit to you for further consideration of publication. Please see more details in our revision. We hope the revised manuscript can satisfy you and meet the high standard requirement for the esteemed Journal. Thank you very much for your attention and consideration.
Reviewer 3 Report
No comment!!
There are many reviews related to this title as follows. No 7 is a recently published review from the author's group. What is worse is 361 words similarity to the No.7 and high suspicious of Plagiarism from iThenticate check.
1. Thermoresponsive Polymers for Biomedical Applications
Polymers 2011, 3(3), 1215-1242. 2. Poly(N-vinylcaprolactam), a comprehensive review on a thermoresponsive polymer becoming popular, Progress in Polymer Science, 53, 2016, 1-51 3. New directions in thermoresponsive polymers, Chem. Soc. Rev., 2013, 42, 7214-7243 4. Thermoresponsive Polymer Colloids for Drug Delivery and Cancer Therapy, Advanced Functional Polymers for Medicine, 11, 1722-1734 (2011) 5. Poly(N-isopropylacrylamide) and Copolymers: A Review on Recent Progresses in Biomedical Applications, Gels 2017, 3, 36; https://doi.org/10.3390/gels3040036 6. Poly(N-isopropylacrylamide) based thermoresponsive polymer brushes for bioseparation, cellular tissue fabrication, and nano actuators
Nano-Structures & Nano-Objects, 16, 2018, 9-23
7. A review on thermoresponsive cell culture systems based on poly(N-isopropylacrylamide) and derivatives. Jiaxing Li, Xiaoguang Fan, Lei Yang, Fei Wang, Jing Zhang & Zhanyong Wang
Int. J. Polym. Mater. Polym.Biomater., 67(6), 371-382 (2018).
Reviewer 4 Report
The manuscript “Preparation and characterization of thermoresponsive poly(N-isopropylacrylamide) for cell culture applications” review the developments of the preparation and characterization of thermoresponsive substrates for cell culture applications. The work was well-done, the manuscript well-organized, and the references are actualized. However, there are several reviews about this topic in the last years and I don´t find the manuscript original enough to be published.
Author Response
Dear Reviewer, we would like to thank you for your careful reading and constructive suggestions. We have revised and polished the manuscript especially in the section of Conclusion and outlook (Revised paper, Page 16, Line 606-646), and now submit to you for further consideration of publication. Please see more details in our revision. We hope the revised manuscript can satisfy you and meet the high standard requirement for the esteemed Journal. Thank you very much for your attention and consideration.
A brief explanation of contribution and originality of this review:
This paper is dedicated to provide a comprehensive review of published articles addressing the preparation and characterization of PNIPAAm based substrates for cell culture applications. It covers the preparation schemes of thermoresponsive planar films and spatial supports, and also includes the characterization techniques specialized for different thermoresponsive substrates. Moreover, recommendations are made for future studies to upgrade the comprehensive properties of thermoresponsive cell culture platforms.
Indeed, several related review articles have been published in the past, but most of them only concern the discussion of a certain issue. However, in accordance with hundreds of relevant literatures and authors’ working experience, this paper presents a comprehensive and accurate summary and comment on the preparation and characterization of thermoresponsive PNIPAAm for cell culture applications.
This review firstly and systematically describes many available fabrication processes for temperature-responsive culture substrates, such as covalent grafting films, physically absorbed coating films, grafted coating films, hydrogels, carriers and scaffolds, and pointed out their respective advantage, disadvantage and adaptive conditions, which provides the initial evaluation for the selection of preparation schemes. Secondly, it gives principal statement of characterization techniques aimed at the above-mentioned thermoresponsive substrates, which proposes referential or usable scheduling plans and measurements for the work of performance evaluation, and aid scholars to make quick and accurate decisions. Finally, this review is concluded with recommendations concerning future work that is needed to address the preparation and characterization of PNIPAAm based substrates for cell culture applications. This systematic and focused review enables readers to quickly locate the required information, and it will be an inspiration for scholars to further proceed and intensify studies on PNIPAAm containing materials.
Round 2
Reviewer 3 Report
This review summarizes the fabrication and characterization methods of thermoresponsive PNIPAAm for the cell culture field, by introducing recent papers in the field. However, the reviewer strongly think that the authors should cite appropriate references, in particular, respecting original works published by pioneers in the fields, including representative reviews.
In the Introduction, [1] should be an original paper which firstly reported the LCST of PNIPAAm. The authors should confirm that [2] reported about the contribution of chemical structure of PNIPAAm on thermoresponsive property. The first application of PNIPAAm in a cell culture dish for safe separation of cell sheets from the culture dish was reported in 1990 (https://doi.org/10.1002/marc.1990.030111109). Its concept was summarized in a review paper (Fabrication of a thermoresponsive cell culture dish: a key technology for cell sheet tissue engineering, Jun Kobayashi and Teruo Okano, 2010 Sci. Technol. Adv. Mater. 11 014111). The reviewer cannot accept Figure 1, which is wrongly described in terms of molecular structure and hydrogen bonding. 1.2. "absorbed" would be "adsorbed". It should be touched how to measure the molecular weight of PNIPAAm polymers which are conjugated or grafted to the thermoresponsive surfaces. Otherwise, it seems difficult to estimate the density of the PNIPAAm on the thermoresponsive surface. The evaluation method of the attachment and detachment of cells should also be described as well as viability of the cells.
Author Response
Dear Reviewer, we would like to thank you for your careful reading and constructive suggestions. We have revised the manuscript in accordance with your excellent advises. The point-to-point replies are listed as followed. Now we submit it to you for further consideration of publication. Please see more details in our revision. We hope the revised manuscript can satisfy you and meet the high standard requirement for the esteemed Journal. Thank you very much for your attention and consideration.
[1] This review summarizes the fabrication and characterization methods of thermoresponsive PNIPAAm for the cell culture field, by introducing recent papers in the field. However, the reviewer strongly think that the authors should cite appropriate, in particular, respecting original works published by pioneers in the fields, including representative reviews.
Reply: Many thanks for your good suggestion. We have cited the related references published by pioneers and representative reviews in the revision according to your advice.
[2] In the Introduction, [1] should be an original paper which firstly reported the LCST of PNIPAAm.
Reply: Many thanks for your professional suggestion. We have cited the original papers which firstly reported the LCST of PNIPAAm, namely new references [1-3] in the revised manuscript.
[3] The authors should confirm that [2] reported about the contribution of chemical structure of PNIPAAm on thermoresponsive property.
Reply: Many thanks for your reminding. Sorry for the mistake. We have deleted reference [2] in the previous manuscript.
[4] The first application of PNIPAAm in a cell culture dish for safe separation of cell sheets from the culture dish was reported in 1990 (https://doi.org/10.1002/marc.1990.030111109). Its concept was summarized in a review paper (Fabrication of a thermoresponsive cell culture dish: a key technology for cell sheet tissue engineering, Jun Kobayashi and Teruo Okano, 2010 Sci. Technol. Adv. Mater. 11 014111).
Reply: Many thanks for your good suggestion. These two references are very important to improve the integrity of this paper.
[5] The reviewer cannot accept Figure 1, which is wrongly described in terms of molecular structure and hydrogen bonding.
Reply: Many thanks for your kind reminding. There are some inaccuracies in Figure 1. We have corrected this. Please see more details in new Figure 1.
[6] 2.1.2. "absorbed" would be "adsorbed".
Reply: Thanks for your reminding. We have corrected this.
[7] It should be touched how to measure the molecular weight of PNIPAAm polymers which are conjugated or grafted to the thermoresponsive surfaces. Otherwise, it seems difficult to estimate the density of the PNIPAAm on the thermoresponsive surface.
Reply: You are absolutely right. The thickness of PNIPAAm tethered layer (related to the molecular weight of PNIPAAm polymers) is the key to acquire the data of grafting density of PNIPAAm modified substrates. Sorry for confusing the two concepts between “grafting density” and “grafting amount”. We have changed the phrases “grafting density” to “grafting amount”.
[8] The evaluation method of the attachment and detachment of cells should also be described as well as viability of the cells.
Reply: Thank you very much for your excellent suggestion which makes us realize we should discuss the evaluation methods of attachment, detachment and viability of targets cells. We have added a new section (3.6) of “Biocompatibility assessment of thermoresponsive substrates”, which includes “cell viability” and “cell attachment and detachment”. Please see more details in our revision.
Round 3
Reviewer 3 Report
The manuscript has been revised in accordance with the reviewer's suggestions and comments, and is acceptable in this journal.